# Evaluation of HLA Region-Specific High-Throughput Sequencing FASTQ Reads Combined with Ensemble HLA-Typing Tools for Rapid and High-Confidence HLA Typing

**DOI:** 10.3390/biology14121717

**Published:** 2025-12-01

**Authors:** Vijay G. Padul, Mini Gill, Jesus A. Perez, Javier J. Lopez, Santosh Kesari, Shashaanka Ashili

**Affiliations:** 1Rhenix Lifesciences, Hyderabad 500038, Telangana, India; 2Krebs Neumann, Cypress, TX 77433, USA; shashi@curescience.org; 3CureScience, 5820 Oberlin Dr., 202, San Diego, CA 92121, USA; 4Medicine, Pan American Cancer Treatment Center, Tijuana 22010, Mexico; 5Neurology, Pan American Cancer Treatment Center, Tijuana 22010, Mexico; 6Asthra Health, Santa Monica, CA 90403, USA; 7The Lundquist Institute, Torrance, CA 90502, USA

**Keywords:** HLA typing, HLA genotyping, HLA high-throughput sequencing, neoantigen peptides, cancer genomics, personalized cancer immunotherapy

## Abstract

Neoantigen peptide vaccine immunotherapy is an emerging therapeutic approach that trains the immune system to recognize cancer cells that produce mutant peptides, thereby inhibiting their growth or eliminating them. Human leukocyte antigens are essential protein molecules that present the mutant peptides on the external surface of cancer cells. Human leukocyte antigens are highly polymorphic within the human population, making accurate genotype identification in individuals particularly challenging. Implementation of neoantigen peptide vaccine immunotherapy requires three types of sequencing data from the patient: DNA exome sequencing from both tumor and normal tissues, and RNA sequencing from tumor tissue. Computational tools for human leukocyte antigen genotyping using high-throughput sequencing data are often not sufficiently accurate when used individually. Employing an ensemble of multiple software tools and sequencing sources can potentially address this limitation, although it typically results in increased processing time. In this study, we evaluated an ensemble method that integrates four human leukocyte antigen-genotyping software tools applied to three sequencing sources from the same individual to achieve high-confidence genotyping. Additionally, we incorporated a human leukocyte antigen region-specific sequencing read-filtering strategy to reduce the processing time required by each software tool, thereby enabling rapid and high-confidence genotyping.

## 1. Introduction

The Human Major Histocompatibility Complex (MHC) plays an essential function in the adaptive immune response by presenting foreign antigens to T lymphocytes [1,2]. The MHC is also known as the human leukocyte antigen (HLA) complex. The HLA genes are located on the p-arm of chromosome 6 at band 6p21.3. HLA genes encode co-dominantly expressed cell-surface proteins that are essential for the recognition of antigens by the immune system [3]. HLA genes are divided into subclasses based on their structure and function. The HLA complex comprises two large clusters of genes, i.e., the MHC class I and MHC class II regions [4,5]. The classical class I genes encoded in the HLA region are *HLA-A*, *HLA-B,* and *HLA-C* [1]. The HLA class I proteins are generally expressed on the outer surface of most somatic cells and their function is to present intracellularly derived peptides to CD8+ cytotoxic T lymphocytes. The HLA class II region encodes *HLA-DPA1*, *HLA-DPB1*, *HLA-DQA1*, *HLA-DQB1*, *HLA-DRA,* and *HLA-DRB1* genes [1]. The HLA class II proteins are generally expressed on the surface of immune cells such as antigen-presenting cells. The HLA class II proteins present peptides of exogenous origin to CD4+ helper T-cells [2,6]. The MHC region is highly polymorphic in the human population. As of October 2025, the IPD-IMGT/HLA database—the official repository for identified HLA alleles—contains 42,996 distinct allelic variants.

HLA class I proteins are known to play a role in tumor immunity [7] as they can present mutated peptides to CD8+ T lymphocytes which may initiate an immune response against cancer cells and can cause death of the cancer cell [8]. An HLA molecule can bind and display a peptide only if the peptide matches the antigen recognition domain of the HLA molecule. Thus, the binding of a peptide to HLA and the subsequent recognition of this complex by a T-cell receptor is highly specific [9]. Therefore, accurate HLA genotyping becomes an essential step in neoantigen peptide-based immunotherapy, which is a cancer antitumor immunotherapy employing specifically designed neoantigen peptide molecules to train the patient’s immune system to recognize cancer cells [10].

Neoantigen peptides are produced in cancer cells as a result of somatic mutations in protein coding genes which render the amino acid sequence different as compared to its natural sequence [11,12]. The mutant proteins may be processed inside the cancer cell and the resultant mutant peptide fragments may be presented to the immune cells on the cancer cell surface in complex with the HLA protein molecule [2]. Recognition of these mutant peptides as non-self by T-cells may further result in a tumor-cell-specific immune response, which may lead to arrested growth or death of the tumor cell [13]. Accurate HLA genotyping ensures optimum design of the neoantigen peptide vaccine for cancer patients [14,15]. However, accurate HLA genotyping is challenging due to high similarity in the alleles of different HLA genes [16] and the exceptional polymorphism found in HLA alleles [6,17].

The HLA-typing methods used for clinical purposes include sequence-specific PCR amplification of HLA gene sequences, which is considered as a gold standard but is labor intensive and expensive [18]. Serological HLA typing is another method that is used; however, the typing resolution is low and the method suffers from a number of other drawbacks [19]. Advances in high-throughput sequencing provide the opportunity to use the whole genome sequence, exome sequence, or RNA-seq sequence data of the individual for HLA genotyping as these data also contain sequences from the HLA region [20,21]. Sequencing of the whole genome or exome of paired tumor–normal tissue and RNA-seq of tumor tissue from the patient is required for the discovery of candidate neoantigen peptides [22,23,24,25]. Among the software tools developed for HLA typing using high-throughput sequencing data, HLA class I-typing algorithms have higher accuracy than HLA class II-typing tools [13,17,26]. The sequencing coverage of the HLA region is found to affect the accuracy of the HLA-typing tools, in addition to the type of algorithm used by the tool [17,26,27,28,29,30]. Due to this limitation, the predicted HLA genotype may be unreliable if a single software and a single sequencing source are used for HLA genotyping. Use of multiple HLA-typing software on multiple data sources from the same individual can provide an opportunity to increase the confidence in the HLA-typing results. In the neoantigen peptide-prediction process pipeline, three sequencing sources from the same patient are used [13] (tumor tissue exome, normal tissue exome, and tumor tissue RNA-seq) and these three sequencing sources can also be useful for HLA typing to increase confidence in the predicted HLA genotype for the patient. In our previous research publication, which was the first study to compare three sequencing sources from the same individual for HLA typing, we analyzed HLA class I genes in cancer tissues through integrative genomic analysis to detect the somatic changes in HLA genes and to identify a high-confidence subset of potentially functional cancer-somatic HLA class I genotypes relevant to personalized neoantigen peptide vaccine immunotherapy (PNPVT) [31]. In that study, we showed that in cancer cells, HLA genes may be affected by somatic changes such as mutation, HLA gene copy loss, or entire chromosome 6 loss, which may impair the HLA function [31]. The same high-throughput sequencing data from the earlier study have been used for the analysis presented in this research article [31].

In this paper, we present an evaluation of HLA typing by using an ensemble of four software tools and HLA region-specific filtered FASTQ reads from the three tissue sources from the same cancer patient for faster and high-confidence HLA typing. The four tools used were OptiType [16], Polysolver [30], ATHLATES [32], and seq2HLA [33]. These four tools utilize different HLA-typing approaches that can be divided into two broad categories: assembly-based methods and alignment-based methods. De novo assembly-based approaches assemble reads into contigs and then align the contigs to the known HLA allele reference sequences [34]. ATHLATES adopts an assembly-based, k-mer pre-filter approach on short read sequences for allele identification and allelic pair inference [32,34]. The other three tools utilize an alignment-based approach, which predicts true alleles based on probabilistic models after aligning high-throughput sequencing reads to the reference HLA sequences [34]. OptiType uses iterative alignment to customized HLA reference sequences [16,34]. Polysolver employs Bayesian alignment to the reference HLA alleles [30,34], and seq2HLA uses iterative alignment to the reference HLA alleles [33,34] for inferring the HLA genotype. ATHLATES and Polysolver use exome sequencing data as input. Seq2HLA uses RNA-seq data as input, while OptiType can use both exome/whole genome and RNA-seq data for HLA typing. We used these software on both exome and RNA-seq sources irrespective of whether the software was developed for use with that data type, with the intention to check if the same HLA allele type may be predicted as with the intended data sources.

The Illumina paired-end sequencing data from three sources—normal tissue exome (blood sample), tumor tissue exome, and tumor tissue RNA-seq–was used for HLA typing using four software tools: ATHLATES, OptiType, Polysolver, and seq2HLA. The most frequent HLA-typing results from these three tissue sources and four software tools were considered as ‘consensus’ two-field HLA allele [35]. The use of four software tools on three FASTQ sequence sources requires high computational power and leads to an increase in the time required to conduct the computational analysis. To overcome this drawback associated with using multiple software tools, we used a subset of FASTQ reads extracted based on successful alignment to known HLA sequences, in order to reduce the number of input FASTQ reads while still aiming to predict the same consensus HLA genotype in less time.

We explored the use of two sets of reference HLA sequence FASTA files for filtering HLA region-specific reads. One reference set, called the All-HLA reference, included all classical and non-classical HLA allele genomic and cDNA FASTA sequences for genotyping both allele types. The other reference set, called the 9-HLA reference, was constructed using only selected HLA class I alleles (*HLA-A*, *HLA-B*, *HLA-C*) and HLA class II alleles (*HLA-DPA1*, *HLA-DPB1*, *HLA-DQA1*, *HLA-DQB1*, *HLA-DRA*, *HLA-DRB1*) specific genomic and cDNA FASTA files, intended for genotyping only these nine HLA alleles. The 9-HLA reference was used for read filtering with the intention of further reducing the number of filtered HLA-specific reads to achieve additional reduction in the time required for HLA typing. The time required for HLA typing using original and filtered FASTQ files was compared. The concordance of HLA genotypes derived from filtered FASTQ reads with those derived from original FASTQ files was estimated to assess whether the same HLA genotypes could be generated by using filtered reads. HLA-typing resolution up to two-fields describes the protein-level amino acid sequence differences encoded in the HLA genes and is the most relevant resolution required for neoantigen peptide immunotherapy [36,37]. Thus, for comparative analysis and to identify the most frequent HLA genotype, all HLA types were converted to two-field resolution.

## 2. Materials and Methods

### 2.1. Samples and Data

Paired-end Illumina sequencing data derived from tumor tissue, normal tissue (blood), and tumor tissue-derived RNA-seq data was obtained from 24 cancer patients and used for the analysis [31]. The data and its underlying quality are described in our earlier publication [31]. All samples were collected with written informed consent and approved by the Institutional Review Board. No alternative HLA-typing information was available for any of the samples included in this study.

### 2.2. Computational Environment

All bioinformatics computational analysis was performed on Ubuntu 20.04.4 wsl on a system with an AMD Ryzen 9 5900HX processor and 64 GB RAM.

A Docker implementation of read filtering and human genome alignment pipeline is provided at https://hub.docker.com/r/rhenixlifesciences/vijaygp-hlareadfilter (accessed on 20 November 2023). A software tool to compile the HLA allele results from multiple software to help in the prioritization of high-frequency HLA alleles is provided at https://github.com/Vijaygpp/HLAnalyze/ (accessed on 24 June 2023).

### 2.3. HLA Database

The latest reference HLA FASTA sequences (version: 3.51.0) were downloaded from the IPD-IMGT/HLA database [6]. These FASTA files were used to construct two reference databases for aligning sample FASTQ reads. The All-HLA reference database was built by combining genomic and cDNA FASTA files for all classical and non-classical HLA alleles. The 9-HLA reference database was assembled by combining genomic and cDNA FASTA files for only nine HLA alleles: *HLA-A*, *HLA-B*, *HLA-C*, *HLA-DPA1*, *HLA-DPB1*, *HLA-DQA1*, *HLA-DQB1*, *HLA-DRA*, and *HLA-DRB1*.

### 2.4. FASTQ Read Filtering

Sample FASTQ reads were aligned to respective reference HLA databases (All-HLA, 9-HLA) using BWA MEM aligner (version: 0.7.17) [38] with default parameters and 14 threads. By default, BWA retains multimapping reads during alignment. The resultant BAM file containing only aligned reads was name sorted using samtools collate (version: 1.15) [39]. Reads were then extracted from the BAM file using samtools fastq (version: 1.15) [39], generating three FASTQ files: two paired-end FASTQ files and one file containing only single-end FASTQ reads. Mate-pair reads corresponding to the single-end FASTQ reads were retrieved from the original paired FASTQ files using seqtk (version: 1.3, https://github.com/lh3/seqtk, accessed on 10 January 2023), and the resulting paired FASTQ files were combined with the initial paired FASTQ output files.

### 2.5. HLA Typing

HLA typing was performed using four tools—ATHLATES, OptiType (version: 1.3.3), Polysolver, and seq2HLA (version: 2.2)—as described in our earlier publication [31]. The HLA-typing resolution provided by ATHLATES and Polysolver was four-field, while for OptiType and seq2HLA it was two-field. For comparative analysis, all the allele types deduced by the four tools were normalized to a two-field resolution. ATHLATES was executed using the Novoalign aligner (version: V3.09.05) (http://novocraft.com/, accessed on 22 April 2022) to map FASTQ reads against the reference HLA database. OptiType [16] was configured with the RazerS3 (version: 3.4) [40] aligner for read alignment, utilizing eight computational threads and employing the CBC solver for Integer Linear Programming (ILP) optimization. Polysolver [30] was executed with the Novoalign aligner (version: V3.09.05) (http://novocraft.com/, accessed on 22 April 2022) to map FASTQ reads extracted from input BAM files to the reference HLA database. Seq2HLA was used with the Bowtie (version: 0.12.7) [41] aligner for read mapping.

### 2.6. BAM Preparation

BAM files were required for HLA typing using Polysolver. The exome FASTQ files were aligned to the human reference genome hg38 using BWA MEM aligner (version: bwa-0.7.17) [38] with default parameters. The resulting BAM files were de-duplicated using Picard MarkDuplicates (version: 2.26.11) (https://broadinstitute.github.io/picard/, accessed on 23 February 2022). RNA-seq data was aligned to the human reference genome hg38 using the STAR aligner (version: 2.7.10a) [42].

## 3. Results

### 3.1. Time Required for FASTQ Read Filtering

The number of reads in the original unfiltered FASTQ files ranged from 26 million to 986 million for exome data and from 33 million to 526 million for RNA-seq data (Figure 1A). After filtering against the All-HLA database, the reads were reduced to 1.3 million–20.3 million for exome data and 189 thousand–15.8 million for RNA-seq data. After filtering against the 9-HLA database, the reads were reduced to 1 million–20 million for exome data and 237 thousand–15 million for RNA-seq data (Figure 1A). The percentage reduction in the number of reads from the original FASTQ file ranged from 2% to 13% for exome data and from 0.4% to 18% for RNA-seq data when filtered against the All-HLA database. Filtering against the 9-HLA database resulted in only a minimal additional percentage reduction in the number of reads (Figure 1B). The time required for FASTQ filtration against the All-HLA database ranged from 16 to 181 min for the exome data and 3 to 113 min for the RNA-seq data. Time required for FASTQ file filtration against the 9-HLA database ranged from 13 to 108 min for exome data and 3 to 73 min for RNA-seq data (Figure 1C). Because Polysolver requires a BAM file as input, original and filtered FASTQ reads were aligned to the human reference genome hg38, and the time required for each alignment was evaluated (Figure 1D).

### 3.2. Time Required for HLA Typing

There was a considerable reduction (4× to 20×) in the time required for HLA genotyping for all four software tools used (Figure 2) when HLA-filtered FASTQ files were utilized. For HLA typing using original exome FASTQ data, OptiType required up to 2040 min, ATHLATES up to 279 min, and seq2HLA up to 444 min. For HLA typing using the original RNA-seq FASTQ data, OptiType required up to 126 min and seq2HLA up to 25 min. For all three software tools, the time required for HLA typing was reduced to a few seconds to a few minutes when All-HLA-filtered FASTQ files were used, with a further slight reduction when 9-HLA-filtered FASTQ files were used (Figure 2A–C). Polysolver uses a human-genome-aligned BAM file as input for HLA typing. The All-HLA-filtered and 9-HLA-filtered reads were aligned to the human reference genome hg38, and the resulting BAM files were used for HLA typing with Polysolver. There was no significant difference at the HLA-calling stage in the time required for HLA typing by Polysolver (Figure 2D). The time reduction for Polysolver occurred at the stage of aligning FASTQ reads to the reference human genome (hg38). Alignment of the original FASTQ files required up to 404 min for exome data and up to 36 min for RNA-seq data. This time was reduced to less than 50 min for exome alignment and less than 5 min for RNA-seq alignment when All-HLA- or 9-HLA-filtered FASTQ files were used (Figure 1D).

### 3.3. HLA Genotyping and Concordance

The HLA class I genotyping results obtained from three sequencing sources in 24 cancer patients using four software tools are shown in Appendix A. The HLA class II genotyping results generated using the two software tools, ATHLATES and seq2HLA, are shown in Appendix A. All distinct two-field HLA calls were represented with unique color shades for visualization in Appendix A. For each HLA gene, the HLA calls derived from the original FASTQ files (bold black font), the All-HLA-filtered FASTQ files (regular black font), and the 9-HLA-filtered FASTQ files (regular brown font) are shown sequentially in Appendix A. The column labels N, T, and R refer to normal exome, tumor exome, and tumor RNA-seq data, respectively. In multiple cases, the same HLA allele type was called by different software using different tissue data sources.

HLA-typing concordance—i.e., the agreement between genotypes obtained using either All-HLA-filtered or 9-HLA-filtered FASTQ reads and those obtained using the original FASTQ reads—was assessed for both HLA class I and HLA class II genotypes across all samples. The patient-level concordance of HLA-filtered FASTQ-derived HLA calls with original FASTQ-derived calls for HLA class I and HLA class II is shown in Figure 3A and Figure 3B, respectively. The patient-level concordance rate for HLA class I alleles ranged from 88% to 100%, and for HLA class II alleles it ranged from 84% to 98%. HLA gene-level concordance is presented in Appendix A for HLA class I and Appendix A for HLA class II. Across samples, the concordance percentage for individual HLA genes ranged from 40% to 100% for all HLA class I and class II genes analyzed. Across the four software tools and three tissue sources, the concordance of HLA class I genotypes with original FASTQ-derived genotypes was 95.97% for All-HLA-filtered FASTQ-derived genotypes and 95.95% for 9-HLA-filtered FASTQ-derived genotypes (Appendix A). For HLA class II genotypes, concordance was 91.63% for All-HLA-filtered FASTQ-derived genotypes and 91.28% for 9-HLA-filtered FASTQ-derived genotypes (Appendix A).

The quality metrics from each HLA-typing tool are superimposed onto the corresponding HLA genotype calls in Appendix A. OptiType and Polysolver do not report explicit quality metrics alongside their final HLA genotype outputs. ATHLATES assigns a score of zero to confidently resolved allelic pairs, whereas nonzero scores indicate partial or incomplete reconstruction of the full-length HLA gene. The majority of HLA class I genotyping results generated by ATHLATES exhibited a score of zero, suggesting high-confidence allele pair assignments and reliable HLA calls (Appendix A). Seq2HLA provides *p* values for each genotype prediction, with most HLA class I results demonstrating statistical significance (*p* < 0.05). To evaluate the impact of HLA region-specific FASTQ read filtering on HLA genotyping quality, comparison of HLA genotype quality metrics was conducted using ATHLATES and Seq2HLA across both HLA class I and class II loci. ATHLATES demonstrated high concordance in quality metrics between HLA-filtered and original FASTQ file-derived HLA genotypes. Specifically, for HLA class I, the All-HLA-filtered reads-derived HLA genotype retained 90.97% of the original reads-derived HLA genotype quality metrics, while the 9-HLA-filtered reads maintained 87%. For HLA class II, the concordance was even higher, with All-HLA- and 9-HLA-filtered reads-derived HLA genotype preserving 98.4% and 98.1% of the original derived HLA genotype quality metrics, respectively. In contrast, Seq2HLA exhibited lower retention of quality metrics when HLA-filtered reads were used. For HLA class II, All-HLA- and 9-HLA-filtered FASTQ-derived HLA genotype retained 84.4% and 83.9% of the original quality metrics, respectively. HLA class I results were comparatively less consistent, with All-HLA- and 9-HLA-filtered reads-derived HLA genotype retaining 74.54% and 74.3% of the original FASTQ-derived genotype quality metrics, respectively. These findings suggest that ATHLATES is more robust to read-filtering strategies, particularly for HLA class II genotyping.

We used the three software tools on non-intended data sources (non-intended tissue source labels highlighted in yellow in Appendix A) in addition to the respective data types for which they were originally designed. Polysolver was developed to be used with exome data and it failed when RNA-seq original FASTQ-reads-aligned BAM files were used for HLA calling, but it worked for 22 patient samples when filtered RNA-seq FASTQ-reads-aligned BAM files were used, producing the same two-field HLA calls as tumor exome data in 87.12% of calls and the same two-field HLA calls as normal exome data in 84.84% of calls (Appendix A). ATHLATES did not work for RNA-seq data from any of the samples (shown as black cells in Appendix A). Seq2HLA is intended for use with RNA-seq data, but it also worked with exome data, producing the same tumor-exome-derived two-field HLA calls as RNA-seq-derived two-field HLA calls for HLA class I in 79.16% calls and for HLA class II in 61.26% of the calls. Using seq2HLA, the same normal-exome-derived two-field HLA calls as RNA-seq-derived two-field HLA calls were produced for HLA class I in 84.04% of calls and for HLA class II in 60.56% of calls. (Appendix A).

### 3.4. Software-Wise Concordance of HLA Genotypes

Of the four software tools used, the highest concordance for HLA class I genotypes was obtained with OptiType, reaching 100% for both All-HLA- and 9-HLA-filtered FASTQ-derived HLA genotypes (Figure 3D, Appendix A). The next highest concordance was observed with Polysolver, at 99.77% for All-HLA- and 99.76% for 9-HLA-filtered FASTQ-derived HLA genotypes (Figure 3E, Appendix A). ATHLATES followed, with concordance of 97.57% for both All-HLA- and 9-HLA-filtered FASTQ-derived HLA genotypes (Figure 3C, Appendix A). For seq2HLA, the HLA class I concordance was 89.58% and 89.35% for All-HLA- and 9-HLA-filtered FASTQ-derived HLA genotypes, respectively (Figure 3F, Appendix A).

For the two software tools used to determine HLA class II genotypes, the highest concordance was achieved with ATHLATES, at 100% for both All-HLA- and 9-HLA-filtered FASTQ-derived HLA genotypes (Figure 3G, Appendix A). For seq2HLA (HLA class II), the concordance was 85.85% and 85.50% for All-HLA- and 9-HLA-filtered FASTQ-derived HLA genotypes, respectively (Figure 3H, Appendix A).

### 3.5. Unanimous and Non-Unanimous HLA Calls

Figure 4 shows the number of HLA-typing results derived from original, 9-HLA-filtered, and All-HLA-filtered FASTQ reads for each sample. Except for HLA class I typing using OptiType (Figure 4D) and HLA class II typing using ATHLATES (Figure 4G), all other software tools showed differences in the number of HLA calls across many samples. Ideally, all four software tools using all three high-throughput sequencing data sources should identify the same two-field HLA allele, and such a unanimous allele call may be considered high confidence. In the present study, single-allele unanimous calls were found for most HLA class I and HLA class II genes (Appendix A) and are shown in green in Appendix A. For HLA class I, the number of identical calls for the same individual across the three data sources and four software tools ranged from a minimum of 2 to a maximum of 11 out of 11 possible calls (Appendix A). For HLA class II, the number of identical calls across the three data sources and two software tools ranged from 2 to 5 out of 5 possible calls (Appendix A). The HLA alleles with a single identical call from all software tools and all three sequencing sources are highlighted in green in Appendix A; these represent high-confidence unanimous HLA calls. For non-unanimous alleles—those for which different HLA types were predicted—the most frequently predicted HLA allele was considered the consensus HLA allele and accepted as the genotype. These consensus alleles are shown in the column labeled ‘C’ in Appendix A for HLA class I and class II, respectively. In cases where predictions tied, the HLA call was deemed ambiguous and is indicated by a white cell in column ‘C’.

To assess the variability of HLA calls made by each software tool across samples, we calculated the ratio of different HLA types called for a given HLA gene for each individual patient. This ratio was obtained by dividing the number of different HLA alleles predicted for that gene locus (by the four software tools using two exome and one RNA-seq data sources for each patient) by the number of patients assessed. Ideally, a ratio of 1 would indicate that all software tools using all sequencing sources predicted a single unanimous HLA allele. The combined results for the four software tools used for HLA class I typing are shown in Appendix A, and those for HLA class II are shown in Appendix A. Software-specific results appear in Appendix A. Among the ratios of different HLA class I type calls (in order: original FASTQ, All-HLA FASTQ, 9-HLA FASTQ), ATHLATES had ratios of 1.08, 1.07, and 1.06; OptiType had ratios of 1.03, 1.03, and 1.03; Polysolver had ratios of 1.12, 1.14, and 1.13; and seq2HLA had ratios of 1.28, 1.31, and 1.31. OptiType showed the best (lowest) ratios (Appendix A). For HLA class II type calls (in order: original FASTQ, All-HLA FASTQ, 9-HLA FASTQ), ATHLATES had ratios of 1.06, 1.06, and 1.06, whereas seq2HLA had ratios of 1.60, 1.49, and 1.48. ATHLATES showed the best ratios (Appendix A).

## 4. Discussion

Computational tools available for HLA genotyping using high-throughput sequencing data are not sufficiently accurate when used individually [13]. This limitation is evident in the variability of HLA-typing results obtained from real-world patient data across three sequencing sources in the present study. While the majority of HLA alleles showed unanimous single-allele calls across software tools and sequencing sources, many alleles still exhibited two or more differing HLA calls. Such discrepancies in HLA allele typing may result from insufficient read coverage for a given allele or limitations in the software algorithms’ ability to accurately resolve the correct HLA type. Generally, the higher the frequency with which a particular HLA allele is predicted across tools and data sources, the greater the confidence in including that allele in the individual’s HLA genotype. In this study, the concordance rate was higher for HLA class I alleles (95.97%) compared to HLA class II alleles (91.63%) (Appendix A). This is consistent with the current status of the lack of adequate accuracy in HLA class II-typing algorithms [13]. The approach proposed in this study to select the most frequently predicted ‘consensus’ HLA alleles across multiple data sources and software tools can enhance confidence in the final HLA genotype for both class I and class II alleles. The advantage of ensemble HLA calling using multiple tools has been demonstrated in previous studies. Li et al. found that the ensemble results of the top three HLA-typing tools were superior to those of individual tools and recommended using concordant alleles integrated from multiple tools to generate reliable HLA results [17]. A study compared ensemble results from eight HLA-typing tools applied to normal whole-exome sequencing data with polymerase chain reaction–sequencing-based typing results and found that HLA genotyping accuracy using high-throughput sequencing data could be improved by combining multiple HLA-typing tools [43]. The ensemble approach may also mitigate tool-specific error patterns and contribute to more reliable HLA-typing results [17].

The results demonstrate that the FASTQ read-filtering approach is effective in significantly reducing the time required for HLA calling, while still predicting the same consensus HLA alleles for HLA class I typing. A reduced FASTQ file size also helps prevent memory-related failures in OptiType, as described by Jian et al. [44], who incorporated a read-filtering strategy in a patch developed for OptiType [44] to address this issue. OptiType uses the RazerS3 [40] aligner, which loads all the FASTQ reads in the memory for performing alignment. This may lead to failure of the HLA typing if FASTQ file size exceeds available memory capacity. Once filtered using the appropriate HLA allele database, FASTQ files can be used with any HLA-typing software, offering the advantage of faster processing. However, this approach should be applied with caution, as it filters only reads that align to known HLA alleles. Novel reads containing HLA sequences not yet included in the IPD-IMGT/HLA database will be excluded. Therefore, it is essential to use the most recently updated IPD-IMGT/HLA allele database for FASTQ read filtering. The FASTQ read-filtering approach is not suitable for discovering novel HLA allele types using de novo approaches [45]. To avoid missing newly added HLA types, the latest version of the IPD-IMGT/HLA database should be used when filtering FASTQ reads for alignment-based HLA-typing software. To achieve complete concordance between HLA genotypes derived from filtered FASTQ files and those obtained from original FASTQ files, improvements in the read filtration process are needed to ensure the full subset of HLA-specific reads is captured. Theoretically, if all HLA-specific FASTQ reads could be accurately isolated, discrepancies in HLA-typing results would be eliminated.

This study focuses on using an HLA region-specific filtered subset of three sequencing data sources from the same individual for HLA genotype prediction. If the true subset of HLA-specific FASTQ reads could be accurately isolated, discordance and mismatches in HLA typing could be minimized. However, some discrepancies between different data sources may still occur. One study applied OptiType to exome data from tumor and normal samples of the same patients and observed distinct HLA-Ia-typing results in a few cases between tumor and control samples [44]. To improve reliability, the authors of that study developed a scoring algorithm that integrated results from both OptiType and Polysolver [44]. Findings from the present study also reveal that mismatches in HLA allele calls can occur even when the same software is applied to three data sources from the same individual. In such cases, the most frequently predicted HLA allele, above a defined threshold, may be considered the genotype for downstream applications such as neoantigen prediction. However, further validation using gold-standard HLA genotyping would be necessary to assess the accuracy of this approach. PCR-based HLA genotypes were not available for any of the samples used in the present study. Lack of comparison with gold-standard HLA genotype is a limitation for this study. In a study conducted by Bauer et al., the concordance of HLA typing by the different variations in PCR-based gold-standard HLA-typing methods was found to be insufficient for clinical applications, leading the authors to recommend an ensemble approach to improve performance [34]. This evidence suggests that even gold-standard HLA-genotyping methods may not yield perfectly accurate results.

The consensus HLA genotype approach may not be suitable for applications in which the entire genotype must be accurately determined, such as HLA genotyping for organ transplantation donor matching. However, for applications where full genotyping is desirable but not strictly necessary—such as neoantigen peptide prediction—this approach can be valuable in identifying a subset of high-confidence HLA alleles. In the absence of reliable HLA genotypes obtained through clinical methods, neoantigen peptides can be predicted using only this subset of high-confidence alleles. The strategy presented in this manuscript is also useful for analyzing legacy Illumina short-read sequencing data, particularly when HLA-typing results from other methods are unavailable or when biological material for laboratory-based analysis is no longer accessible.

## 5. Conclusions

The results demonstrate the utility of the described HLA-genotyping approach for achieving faster and high-confidence two-field HLA typing. Incorporating multiple FASTQ data sources from same individual alongside multiple HLA-typing software tools can enhance confidence in the predicted HLA genotype. Additionally, the FASTQ read-filtering strategy significantly reduces the time required to complete HLA genotyping analysis. Further refinement of the HLA-specific read extraction process could improve the accuracy and broaden the applicability of this approach.

## Figures and Tables

**Figure 1 biology-14-01717-f001:**
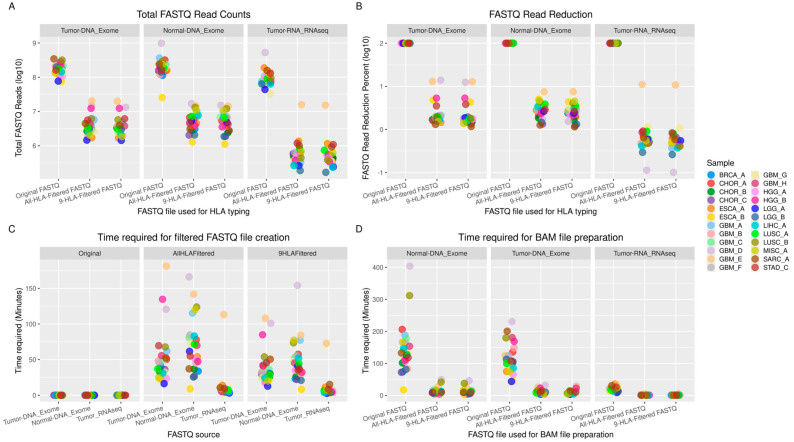
(**A**) Effect of read filtering on FASTQ read count. (**B**) Percent read reduction (log10) achieved after read filtering. (**C**) Time required for filtered FASTQ file creation. (**D**) Time required for BAM file creation using original FASTQ, All-HLA-filtered FASTQ, and 9-HLA-filtered FASTQ.

**Figure 2 biology-14-01717-f002:**
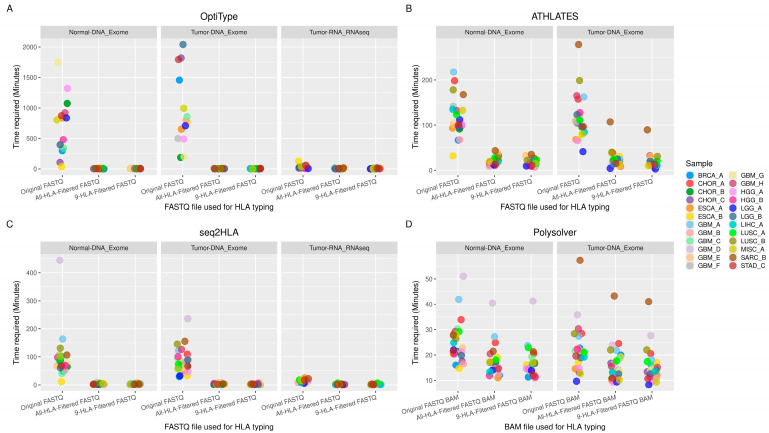
Time required for HLA typing using original FASTQ, All-HLA-filtered FASTQ, and 9-HLA-filtered FASTQ (from Normal-DNA exome, Tumor-DNA exome and Tumor-RNA RNA-seq data) using four software tools. (**A**) ATHLATES, (**B**) OptiType, (**C**) seq2HLA, and (**D**) Polysolver.

**Figure 3 biology-14-01717-f003:**
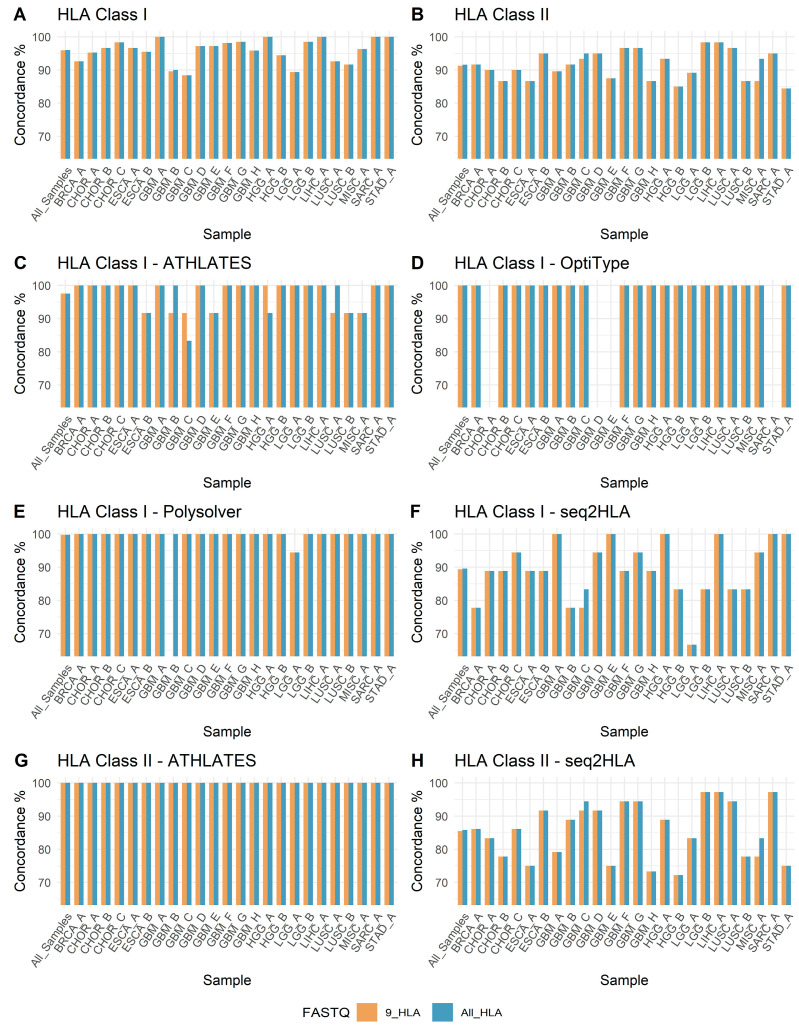
Concordance of HLA-typing results derived from HLA-filtered FASTQ reads (9-HLA- and All-HLA-filtered) with those derived from original FASTQ reads.

**Figure 4 biology-14-01717-f004:**
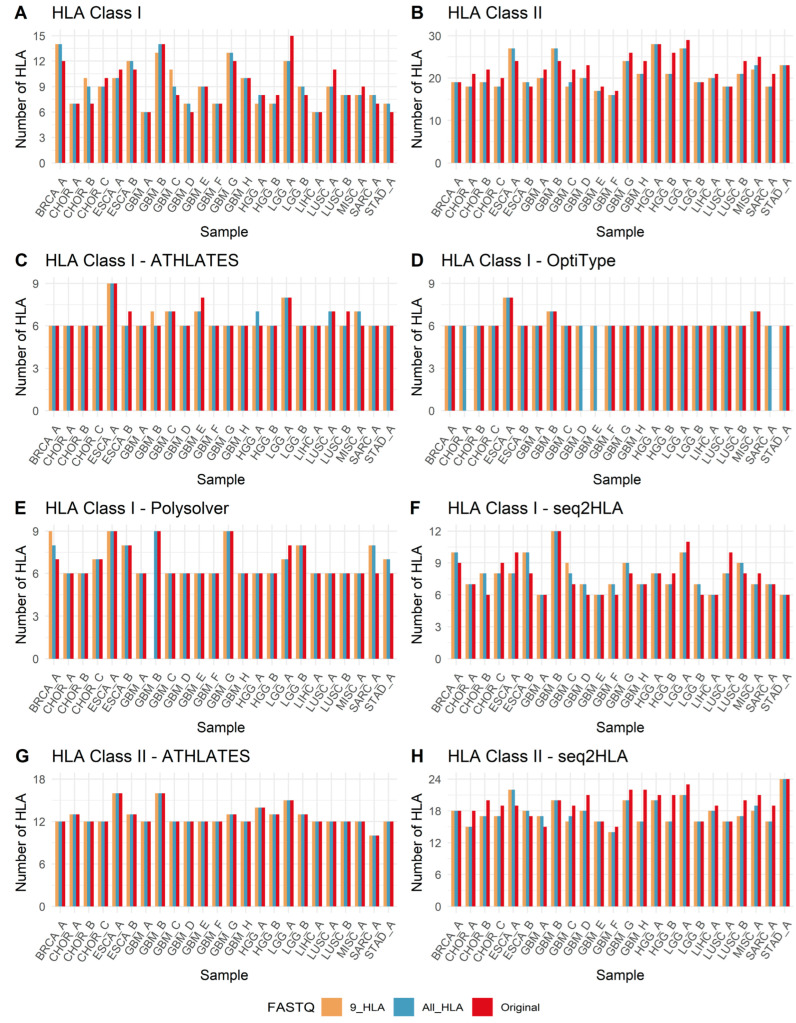
Number of HLA-typing results derived from original, 9-HLA-filtered and All-HLA-filtered FASTQ reads from each sample.

## Data Availability

The data that support the findings of this study are available on request from the corresponding author. The data are not publicly available due to privacy or ethical restrictions.

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
