# Peer review of "Evaluation of HLA Region-Specific High-Throughput Sequencing FASTQ Reads Combined with Ensemble HLA-Typing Tools for Rapid and High-Confidence HLA Typing"

_biology, 2025, doi:10.3390/biology14121717_

Round 1
Reviewer 1 Report
Comments and Suggestions for Authors
In the manuscript “Evaluation of Use of HLA Region Specific High Throughput 2 Sequencing FASTQ Reads Combined with Ensemble HLA 3 Typing Tools for Faster and Accurate HLA Typing”, the authors present and validate an ensemble approach to HLA genotyping from sequencing data. The topic is of importance to many areas of research, although the novelty of this manuscript is primarily derived from the combination of existing tools. There are a number of questions or enhancements that would improve the manuscript prior to publication.
Major concerns:
- One of the innovations of this study is the use of a prefilter step to subset genome or exome reads to those mapping to HLA. This is executed by mapping with BWA-mem against a reference comprised of HLA sequences. A potential concern is that multimapping reads (which will almost certainly occur against the HLA database) could be dropped and under-counted. Can the authors explicitly quantify this? One option could be to use a different aligner, such as bowtie2, which has more customization over handling of ambiguous results, and to force the aligner to retain any alignment. If the total reads retained by some other strategy is comparable to BWA-mem, this would be sufficient. Alternately, if the authors could quantify the multimapping rate produced by BWA-mem and verify such reads are handled appropriately, this would also be sufficient.
- The level of analysis of the consensus results (Figures 3-6) is well below what I would expect. While the authors find agreement in HLA genotypes in many samples, a non-trivial number of samples are discordant. Their strategy would be of much greater use if they inspected datasets with genotyping disagreements to better understand the cause. Thanks to the filtering step used by the authors, it should be tractable to inspect the HLA-specific reads for specific datasets.
- The read depth after filtration varied by a log – does this impact HLA typing?
Minor concerns:
- Figures 3 and 4 are quite large and might be more appropriate as supplemental data.
- While Figures 5 and 6 provide a summary relative to Figs 3-4, they are still quite large. It would be helpful to provide a higher level summary of accuracy, perhaps a graph based on the bottom two rows of Fig 5A-B and Fig 6A-B.
- The authors perform genotyping using three different donor datasets. Would there be value in pooling the passing reads (which might provide the greatest amount of information) as opposed to analyzing each dataset separately?
- The genomic datasets should contain intronic sequence, while the RNA-seq would not. Can the authors comment on the implications of this for the various HLA typing tools?
Author Response
Response to Reviewer 1:
In the manuscript “Evaluation of Use of HLA Region Specific High Throughput 2 Sequencing FASTQ Reads Combined with Ensemble HLA 3 Typing Tools for Faster and Accurate HLA Typing”, the authors present and validate an ensemble approach to HLA genotyping from sequencing data. The topic is of importance to many areas of research, although the novelty of this manuscript is primarily derived from the combination of existing tools. There are a number of questions or enhancements that would improve the manuscript prior to publication.
Major concerns:
Comments 1: One of the innovations of this study is the use of a prefilter step to subset genome or exome reads to those mapping to HLA. This is executed by mapping with BWA-mem against a reference comprised of HLA sequences. A potential concern is that multimapping reads (which will almost certainly occur against the HLA database) could be dropped and under-counted. Can the authors explicitly quantify this? One option could be to use a different aligner, such as bowtie2, which has more customization over handling of ambiguous results, and to force the aligner to retain any alignment. If the total reads retained by some other strategy is comparable to BWA-mem, this would be sufficient. Alternately, if the authors could quantify the multimapping rate produced by BWA-mem and verify such reads are handled appropriately, this would also be sufficient.
Response 1:
We thank the respected reviewer for reviewing the manuscript and providing valuable suggestions. We quantified multimapping reads using two software i.e. BWA-mem and STAR aligner, the results are presented in Table R1.
|
Software |
Sample |
Total input FASTQ reads |
HLA filtered FASTQ reads |
Filtered % |
Uniquely mapped |
Multimapped |
Multimapped % (out of filtered FASTQ reads) |
Multimapped & Uniquely mapped identification method |
|
BWA |
Normal |
363109122 |
7244226 |
2.00 |
NA |
607259 |
8.38 |
count 'XA:Z:' tags |
|
Tumor |
272756806 |
4444631 |
1.63 |
NA |
466885 |
10.50 |
count 'XA:Z:' tags |
|
|
RNA |
76321774 |
243844 |
0.32 |
NA |
11439 |
4.69 |
count 'XA:Z:' tags |
|
|
STAR |
Normal |
363109122 |
10291774 |
2.83 |
50828 |
394594 |
3.83 |
STAR Log.final.out file |
|
Tumor |
272756806 |
5392628 |
1.98 |
28138 |
207438 |
3.85 |
STAR Log.final.out file |
|
|
RNA |
76321774 |
1390146 |
1.82 |
12108 |
47814 |
3.44 |
STAR Log.final.out file |
Table R1: Quantification of multimapping reads
BWA-mem does not drop multimapping reads by default. In the bam files produced by BWA-mem, the multimapping reads are tagged with “XA” tag which enlists alternative locations to which the read aligns. Following is the figure for your reference (Figure R1).
![]() |
|
Figure R1: BWA-mem aligned bam file showing multimapping reads tagged with “XA” tag. |
STAR aligner reports multimapping reads number as well as uniquely mapped reads number in the final report produced along with the bam file.
The quantification from these two sources on three samples is shown in Table R1. Both the software have retained multimapping reads but the retained numbers are different due to the algorithm utilized by each software. We experienced that the STAR aligner is much slower than BWA-mem for reads filtering step. We also tried to use bowtie2 but it was found to be extremely slow to align against HLA reference thus rendering it impractical for the purpose.
Comments 2: The level of analysis of the consensus results (Figures 3-6) is well below what I would expect. While the authors find agreement in HLA genotypes in many samples, a non-trivial number of samples are discordant. Their strategy would be of much greater use if they inspected datasets with genotyping disagreements to better understand the cause. Thanks to the filtering step used by the authors, it should be tractable to inspect the HLA-specific reads for specific datasets.
Response 2:
We agree with the respected reviewer that understanding the cause for the discordance in the HLA calls would be helpful to increase the utility of the filtering approach. At this stage we could not determine the exact causes for the genotype discordance for specific samples. We theorize that one of the major cause could be the lack of extraction of all HLA specific reads in the filtered FASTQ subset. Attempting to establish exact cause for each discordance event would require extensive analysis and thus would require a lot of time. To achieve this, extraction of reads used by each HLA typing software and their comparison with HLA reference filtered FASTQ subset will be required. Considering multiple sequencing sources and multiple software used, this would require an extensive analysis effort. This could better serve as a topic for new research article where the issues about extraction of all HLA specific reads could be studied.
Comments 3: The read depth after filtration varied by a log – does this impact HLA typing?
Response 3:
The original exome unfiltered FASTQ file contains reads that represent sequences from ~20,000 coding genes and the original RNA-seq unfiltered FASTQ file contains tissue specific mRNA expression profile specific FASTQ reads. HLA specific FASTQ reads constitute a small subset of the total FASTQ reads. Thus quantification difference by a log is expected. This difference should not affect HLA typing as the reads that do not originate from the HLA region are filtered out and excluded by HLA typing software. The read filtration should ideally extract only HLA specific reads. If perfectly achieved, the filtration process should not affect HLA typing results. In real world scenario the read filtration may affect HLA typing if some HLA specific reads are missed during read filtration step.
Minor concerns:
Comments 4: Figures 3 and 4 are quite large and might be more appropriate as supplemental data.
Response 4:
We agree with the respected reviewer. We have moved Figures 3 and 4 to supplementary section as Supplementary figures S1 and S2 respectively.
Comments 5: While Figures 5 and 6 provide a summary relative to Figs 3-4, they are still quite large. It would be helpful to provide a higher level summary of accuracy, perhaps a graph based on the bottom two rows of Fig 5A-B and Fig 6A-B.
Response 5:
As per the suggestion by the respected reviewer, two new figures (Figures 3 and 4) have been created to provide a higher level summary. The two figures are based on the bottom two rows of Figure 5A-B and Figure 6A-B and also based on software specific Supplementary Figures S5A-B to S10A-B. The new Figure 3 provides bar plots of HLA typing concordance of HLA typing results based on All-HLA and 9-HLA filtered reads with that of original reads based HLA typing for all 24 samples. The Figure 4 provides bar plot comparisons of number of HLA called using the three FASTQ files for each of the 24 patient’s samples.
Comments 6: The authors perform genotyping using three different donor datasets. Would there be value in pooling the passing reads (which might provide the greatest amount of information) as opposed to analyzing each dataset separately?
Response 6:
We thank the respected reviewer for this interesting suggestion. Certainly, pooling reads from three sources would help to increase the accuracy of HLA typing. The pooling would increase the coverage of HLA specific reads which would enhance the statistical confidence in the HLA typing results. The HLA typing software perform better with increased FASTQ reads coverage. There could also be certain pitfalls to consider. As two of the sequencing sources are derived from tumor tissues (Tumor exome and tumor RNA-seq), the somatic changes in the cancer tissue might deviate the HLA typing results due to somatic mutations or chromosome 6 copy number alterations. This might affect the HLA typing results. Another pitfall could occur if the three samples are not originating from single individual (due to laboratory mislabeling etc.). To avoid this, confirmation of origin of tissue data would be required. This can be achieved by carrying out the independent HLA typing from three individual samples and pooling the reads from three samples to perform the pooled HLA typing. Implementation of this strategy would also require comparison of tumor samples where HLA region is affected and the samples in which the HLA region is unaffected. In addition, extraction of complete true subset of HLA specific reads will also be essential. This could be carried out as a separate study which we will undertake in the future.
Comments 7: The genomic datasets should contain intronic sequence, while the RNA-seq would not. Can the authors comment on the implications of this for the various HLA typing tools?
Response 7:
Exome sequencing data originates from DNA coding regions. Though these are enriched for coding regions (exons) the reads may overlap with surrounding intronic regions. The reads alignment software used by HLA typing tools specific for exonic or genomic data are specialized to handle these reads and align them properly at the respective genomic locations. In the case of HLA reference, these reads will primarily align to HLA genomic FASTA reference sequences which includes both intronic and exonic sequences.
In case of FASTQ reads arising from RNA-seq experiments, the reads may or may not contain the intronic sequences, depending on the RNA capture at pre-mRNA or mature mRNA stage. RNA-seq aligners used by RNA specific HLA typing tools can handle alignment of these reads as these aligners are splice aware and can split the RNA reads spanning exon splice junctions at the HLA genomic FASTA reference sequences. Alternatively HLA cDNA sequences are also included in the HLA reference where spliced FASTQ reads can be aligned without need to split the splice junctions.
Reviewer 2 Report
Comments and Suggestions for Authors
Interesting work. Two aspects are missing. I would expect to see concordance results when long range sequencing is used especially using unknown samples instead of short range seuencing with known samples. That means it would be helpful to see the power of the methodology when unknown samples are typed using the tow abobe mentioned methods. Currently, the laboratories switch or alreqady switch from NGS to ONT. A comparison would be very helpfull.
Comments on the Quality of English LanguageThe simple abstract needs polishing especially when HLA is reported and presented. Perhaps a HLA specialist might help. The main text is quite difficult to read. A polishing here might also help. Making use of fewer Figures might help.
Author Response
Response to Reviewer 2:
Comments 1: Interesting work. Two aspects are missing. I would expect to see concordance results when long range sequencing is used especially using unknown samples instead of short range seuencing with known samples. That means it would be helpful to see the power of the methodology when unknown samples are typed using the tow abobe mentioned methods. Currently, the laboratories switch or alreqady switch from NGS to ONT. A comparison would be very helpfull.
Response 1:
We thank the respected reviewer for reviewing the manuscript and providing valuable suggestions. Long read sequencing has many advantages for HLA typing as compared to short reads sequencing. It would certainly be interesting to compare the concordance results derived from Illumina short reads sequencing and ONT long read sequencing. Unfortunately, for the samples studied in the current manuscript, there are no biological material or tissue samples available to perform long read sequencing analysis. In addition, a comparison of long read and short reads based HLA typing with clinical PCR based and serological methods would also be helpful to determine the accuracy of these methods. We are planning to include these analyses in the future study. The strategy presented in the current manuscript would also be helpful to analyze legacy Illumina short read sequencing data for which HLA typing results from other methods are not available and biological material for laboratory analysis are also not available (Page 13, line 466).
Comments on the Quality of English Language
Comments 2: The simple abstract needs polishing especially when HLA is reported and presented. Perhaps a HLA specialist might help. The main text is quite difficult to read. A polishing here might also help. Making use of fewer Figures might help.
Response 2:
We thank the respected reviewer for the valuable suggestion. The simple abstract has been revised and formatting issues with HLA names has been corrected. We have corrected the HLA nomenclature. We have also revised the main text to improve the language and grammar. We hope the respected reviewer would find the revised version acceptable.
In the revised manuscript the number of figures from the main text have been reduced from six to four, including two new figures (Figure 3 and 4) prepared as per the suggestion of another reviewer. The four figures from initial submission (Figures 3 to 6) have been moved to supplementary section.
Reviewer 3 Report
Comments and Suggestions for Authors
The authors introduced a rapid and reliable testing method suitable for accurate HLA genotyping analysis from tumor and normal tissues, utilizing multiple FASTQ data sources and multiple HLA typing software.
While there are no major changes to this paper, please explain the following points.
- The authors stated that the HLA typing introduced in this paper is suitable for discovering novel antigen peptides. If you have any results showing that novel antigen peptides can be predicted from the sequence data obtained from the 24 cancer patients tested, please present specific examples.
- This paper analyzes sequence data obtained from cancer and normal tissues, but why is sequence data from PBLs not used? If the purpose is accurate HLA typing from tissues, it might be better to change the title.
- Is Figure 2D on page 6, line 250, actually Figure 2C?
- Should Figure 54 on page 7, line 260, be Figure 4 instead?
- What criteria were used to select the HLA typing results presented in Figures 5 and 6 from each cancer patient's specimen?
Author Response
Response to Reviewer 3:
The authors introduced a rapid and reliable testing method suitable for accurate HLA genotyping analysis from tumor and normal tissues, utilizing multiple FASTQ data sources and multiple HLA typing software.
While there are no major changes to this paper, please explain the following points.
Comments 1: The authors stated that the HLA typing introduced in this paper is suitable for discovering novel antigen peptides. If you have any results showing that novel antigen peptides can be predicted from the sequence data obtained from the 24 cancer patients tested, please present specific examples.
Response 1:
We thank the respected reviewer for reviewing the manuscript and providing valuable suggestions. We wish to clarify the misunderstanding. The manuscript does not claim that the method introduced in the paper is suitable for discovering novel antigen peptides. The proposed method is aimed at identification of high confidence subset of the patient HLA genotype and reduction of the time required for HLA genotyping of cancer patients who are planned to undergo neoantigen peptide immunotherapy. The discovery of the novel antigen peptides is the step to be performed after the HLA genotyping is achieved, and is not the part of current manuscript. The therapeutic neoantigen peptide design is dependent on accurate HLA genotyping (Page 2, line 78). The current understanding and suggestion in the manuscript is that accurate HLA genotyping will be essential for designing best neoantigen peptide sequences which may ultimately lead to best therapeutic outcome.
Neoantigen peptide design is a separate area of study which focuses on design optimization of peptide sequences that can potentially bind to HLA protein molecules. Neoantigen peptide design is not a part of the current manuscript. To estimate HLA-peptide binding affinity, a number of open source in silico peptide design software for the prediction of HLA-peptide binding affinity (MHCflurry, NetChop, NetMHC etc.) and pipelines for discovery of neoantigen peptides (pVACtools, OpenVax etc.) have been developed.
Comments 2: This paper analyzes sequence data obtained from cancer and normal tissues, but why is sequence data from PBLs not used? If the purpose is accurate HLA typing from tissues, it might be better to change the title.
Response 2:
We would like to point out to the respected reviewer that all the normal tissue sequence data used in the study is derived from PBLs. All Normal samples used in the current study are blood sample PBL. As mentioned in the original submission text page 3 line 133 “three sources such as blood exome, tumor exome and tumor RNA-seq was used”. We have modified the text in the Materials and methods section of the revised manuscript to specify that blood was used as normal tissue sample (revised manuscript: page 4, line 145 & line 175).
Comments 3: Is Figure 2D on page 6, line 250, actually Figure 2C?
Response 3:
We thank the respected reviewer for pointing out this mistake. The figure referred to is actually Figure 1D. We have corrected the text in the revised manuscript. (Page 6, line 257).
Comments 4: Should Figure 54 on page 7, line 260, be Figure 4 instead?
Response 4:
We thank the respected reviewer for pointing out this mistake. The figure referred to was Figure 4 in the original submission. In the revised version the same figure is as Supplementary figure S2. We have corrected the text in the revised manuscript. (Page 7, line 267).
Comments 5: What criteria were used to select the HLA typing results presented in Figures 5 and 6 from each cancer patient's specimen?
Response 5:
In Figures 5 and 6 of the original manuscript, all the HLA typing results from four software and three sequencing sources were presented. There were no selection criteria. In the best case, unanimous HLA typing results, represented by “1” and highlighted in green background are the high confidence unanimous results. Any number other than “1” represents disagreement of HLA typing among software or sample source. In the revised manuscript, Figures 5 and 6 have been moved as Supplementary Figures S3 and S4.